# Implementation of Bio-Risk Management System in a National Clinical and Medical Referral Centre Laboratories

**DOI:** 10.3390/ijerph18052308

**Published:** 2021-02-26

**Authors:** Fatma Lestari, Abdul Kadir, Thariq Miswary, Cynthia Febrina Maharani, Anom Bowolaksono, Debby Paramitasari

**Affiliations:** 1Occupational Health and Safety Department, Faculty of Public Health, Universitas Indonesia, Depok West Java 16424, Indonesia; abdulkadirindo22@gmail.com; 2Disaster Risk Reduction Centre (DRRC), Universitas Indonesia, Depok West Java 16424, Indonesia; alaksono@sci.ui.ac.id (A.B.); debbyparamitasari@gmail.com (D.P.); 3National Clinical and Medical Referral Centre Laboratories, Bekasi 17111, Indonesia; thariqmiswary@gmail.com; 4Occupational and Environmental Health Department, Public Health Faculty, University of Iowa, Iowa City, IA 52242, USA; cynthia-maharani@uiowa.edu; 5Cellular and Molecular Mechanisms in Biological System (CEMBIOS) Research Group, Department of Biology, Faculty of Mathematics and Natural Sciences, Universitas Indonesia, Depok West Java 16424, Indonesia

**Keywords:** bio-risk, management, implementation, laboratory, gap analysis, ISO 35001:2019

## Abstract

The increasing threats from biological agents have become a concern in laboratories, and emerging infectious diseases have demanded increased awareness and preparedness of laboratory facilities. Bio-risk assessment is needed to provide a framework for organisations to establish a comprehensive bio-risk management system. The assessment criteria should include both biosafety and biosecurity measures. Laboratories in Indonesia play a significant role in public health interventions in term of disease screening, diagnosis and medical decision making. The National Clinical and Medical Referral Centre Laboratories have the potential of daily exposures to dangerous biological materials. This study aims to identify the gap between bio-risk management system implementation and International Standard Organisation (ISO) 35001:2019 requirements. The 202 items in ISO 35001:2019 are categorized into seven main elements. The findings show that more than half of the elements on ISO 35001:2019 have been implemented in these centres. Good performance was identified at lab 4 and 5 which obtained the highest scores, particularly in the context of organisation, planning, operation and improvement elements. However, the widest gap was found in leadership, support and performance evaluation. One way to address this would be to create written rules and regulations at the laboratory top management level to require all laboratory facilities to comply to the bio-risk policies, rules, and regulations.

## 1. Introduction

A laboratory service facility has the function to extract biological materials which are then managed globally for certain purposes like education, scientific, pharmaceutical and health-related production. These facilities are also potentially exposed to dangerous microorganisms that will harm people and disrupt the environment [1]. Hence, laboratories that handle hazardous pathogens must be responsible for managing the safety and security of the laboratory against the threats released by these biological agents [2,3]. Bio-risk management as a guideline to regulate assets and manage permissions to biological materials requires a comprehensive system to be implemented, which includes policy and management aspects [4]. This comprehensive system aims to provide a framework for organisations to conduct bio-risk assessment and ensure that all laboratory’s activities can be sustainably maintained [5].

In a global setting, many countries have implemented bio-risk management for laboratories that encompasses bio-safety and bio-security aspects. A study in Kenya presented enhanced training for laboratory staffs on bio-security and bio-safety as highly associated with compliance to bio-risk codes [6]. However, a study in Africa showed that most laboratories are having issues with undeveloped transmission control activities due to lack of awareness, inadequate number of trained personnel in infection control, inadequate infrastructure and procurement barriers that result in poor infection control [7,8]. Lower bio-safety precautions are linked with poor personal protective behaviour among personnel handling clinical samples, who are at a higher risk of being exposed by infectious agents [9,10]. In bio-risk management assessment, complying to biosafety requirements requires provision of appropriate protective equipment for these laboratory personnel [11,12,13].

A bio-safety audit conducted in two university laboratories in Singapore indicated that inconsistency of bio-safety commitment affects procedures in a laboratory [14]. In addition to providing personal protective equipment, there should be a combination of policies and systems to protect laboratory workers from intentional or unintentional risk of infections and human errors, as well as appropriate steps for biomedical waste management [4,5]. A documented standard operational procedure (SOP) is an important part of this effort to ensure the consistency of test performance and reliable outcomes. Written SOPs should also be accompanied by a comprehensive strategy for increasing and sustaining safe laboratory behaviours which involve staff in the development and compliance to the final SOPs and ensuring that all administrative controls are followed [15,16].

As in other countries, laboratories in Indonesia also play a fundamental role in public health interventions in term of disease screening, diagnosis and medical decision making. Laboratory facility as the forefront to detect disease is supposed to be a safe and secure workplace for their workers. In the current coronavirus disease 2019 (COVID-19) pandemic, the rapid spread of this virus has become an alert for all laboratories to increase their capabilities in providing fast and precise diagnostic supports as the load of samples from suspected patients is increasing on daily basis [17,18]. In order to improve their capabilities, laboratories need skilled human resources, good microbiological techniques, adequate protective clothing, adequate personal protective equipment and decent laboratory equipment and maintenance [19]. Regular monitoring and assessment of biosafety will not only promote a safer working environment, but also affect the laboratory’s quality service [14].

A simple tool checklist has been developed by the World Health Organisation (WHO) in its Laboratory Biosafety Guideline. The National University of Singapore (NUS) also published a Bio-risk Management Manual to assess laboratories in educational settings. Bio-risk assessment in a laboratory can be measured using various tools or combinations based on its purposes. However, the authors chose to use the International Standard Organisation (ISO) 35001:2019 because only a few studies use this tool despite the fact that this tool is the international standard assessment for bio-risk management in laboratories. In addition, ISO 35001:2019 is commonly used to identify, assess, control and monitor the risks linked with biological materials. The tool also includes several aspects to be assessed, such as the context of organisation, leadership, support, planning, operation, performance evaluation and improvement [20]. Hence, this study aims to identify the gap between bio-risk management system implementation and requirements in the ISO 35001:2019 at a national referral laboratory centre [21]. The National Clinical and Medical Referral Centre Laboratories in Indonesia have a potential of daily exposure to dangerous biological materials during sampling activities in the laboratory. Thus, this laboratory provides an excellent opportunity to analyse the implementation of bio-risk management and to decide the measure controls that can be replicated in all clinical and medical laboratories.

## 2. Materials and Methods

### 2.1. Study Designs

A descriptive method was used in this study to examine the implementation of the Bio-risk Management System in a National Clinical and Medical Referral Centre Laboratories, Indonesia by combining the qualitative and quantitative approach. A gap analysis was also performed based on results, which is the total number of items that had not been implemented from the 202 items on the checklist. The attainment of each laboratory would be categorised as good if the score obtained reached a total value of ≥50% [22]. The assessment was performed by a Professor in Occupational Health and certified safety and biological expert in bio-risk management systems. The primary data were collected through direct observation, bio-risk management instrument checklist, and interviews with the head of laboratories. Secondary data as well as literature studies were also used to complement the gathered information. The theoretical concept used in this study is shown in Figure 1 below.

### 2.2. Subjects Study

This study was carried out at the National Clinical and Medical Referral Centre Laboratories, Indonesia. This centre is the biggest private laboratory chain which consist of 174 branches, 3 regional referral laboratories, and one national referral laboratory centre across Indonesia. The national referral laboratory centre which is located in Central Jakarta has been established since 2008. This referral laboratory which occupies a high-rise building with 10 stories and a basement has become the nationwide referral centre for clinical specimen assessment and collects samples from all branches and public hospitals as well as private hospitals in Indonesia. This centre offers more than 600 types of laboratory’s diagnostic tests with the average services capacity of 130,000 tests each month.

The centre only offers specific services to assess biological sample from the human body. The samples accepted by this centre are red blood samples, body fluids, and also body tissues. Several materials are used in the laboratory processes such as chemical reagents and formalin to preserve biological specimen. Some equipment are usually used in the laboratory such as a biosafety cabinet (BSC), autoclave, centrifuge, fume hood and automatic analysis equipment. The HVAC (heating, ventilation and air-conditioning) system of the National Clinical and Medical Referral Centre’s building consists of split duct AC. This system allows temperature and humidity control, as well as pathogen contamination protection in the room. There is no air circulation between laboratories and offices.

This study used the total sampling technique with the consideration to include all laboratories that utilize biological sampling during their activities. With this inclusion criteria, eight main laboratories were selected as sample population at National Clinical and Medical Referral Centre Laboratories (Table 1). This study was performed from May to June 2020. 

### 2.3. Instruments and Data Analysis

The checklist applied in this study was the ISO 35001:2019 checklist (Bio-risk management for laboratories and other related organisations). The license for using this tool was accepted by the authors in line with the ISO store order OP-437087. The framework of ISO 35001:2019 is based on a management system known as the plan-do-check-act (PDCA) principle. In addition, ISO 35001:2019 point outs the scope of this document by defining the process to identify, assess, control and monitor the risks of hazardous biological materials. This international standard consists of 10 clauses: (1) scope, (2) normative references, (3) terms and definition, (4) context of the organisation, (5) leadership, (6) planning, (7) support, (8) operation, (9) performance evaluation and (10) improvement [20]. According to these clauses, there were 202 checklist items to support the main clauses of ISO 35001:2019, namely clauses 4 to 10 (Table 2). Data processing was done by computerization and data obtained were then processed by calculating each answer. Three categories were then used to describe the condition of the laboratory: Score 0.0 reflected the absence of documentation and evidence; score 0.5 referred to the presence of an implementation and evidence; and score 1 was given for the implementation of bio-risk management with documentation and evidence. The implementation and gap analysis scores were calculated according to formula below:(1)$$Total\;implementation\;score\;\left(T\right)=\frac{total\;aspect\;implemented\;\times \;\;100}{total\;aspect\;implemented+total\;aspect\;not\;implemented}$$

### 2.4. Ethical Considerations

The study went through the ethical assessment procedure and was approved by the Research and Community Engagement Ethical Committee of Faculty of Public Health Universitas Indonesia under approval letter number 170/UN2/F10/D11/PPM.00.02/2020.

## 3. Results

The National Clinical and Medical Referral Centre Laboratories have eight laboratories, including pathology anatomy, haematology, cytogenetic, molecular, microbiology, urinalysis, automation and immuno-serum laboratories. Six of the laboratories were categorized as BSL 1 while one was BSL 2 and the other was BSL3 with high containment available (Table 1). In addition, National Clinical and Medical Referral Centre Laboratories have already been certified for ISO 9001:2015 regarding Quality management system and the National Standard Indonesia ISO 15189 for Clinical/Medical Laboratory, Occupational Safety and Health Management Systems (OHSMS) based on Government Regulation No. 50 of 2012 This centre was also certified for OHSAS 1800 about Occupational Health and Safety Assessment Series and also had CAP Certification (College of American Pathologists).

Based on the checklist used, we observed seven main clauses that consisted of different items for each clause, starting from the context of the organisations to improvement (Table 2). The achievement analysis was based on points applied to each laboratory from the 202 questions adopted from the ISO 35001:2019. Overall, 8 laboratories of the subject study had obtained an implementation score of greater than 50%, with an average score of 84%. In more detail, the score for each laboratory was 83.9% and 84.4% for laboratory 1, 2, 6, 7, 8 and laboratory 4 and 5, respectively. The score was shown to be relatively similar for each laboratory at the national referral centre. The company had previously implemented a quality management system, namely ISO 9001:2015 for occupational health and safety management system, and had accreditation by the national accreditation committee SNI ISO 15189. However, there was around 16% of the total items in the checklist that had not been implemented. 

Figure 2 presents the implementation scores for bio-risk management according to each element of ISO 35001:2019, starting from context of organisation to improvement. All elements obtained more than a total value of 50% with an average score as follows: context of organisation (64.3%), leadership (75%), planning (90.9%), support (82.7%), operation (93%), performance evaluation (87%) and improvement (92.3%). Table 3. illustrates the total score and the percentage of implementation of bio-risk management per sub-clauses. It can be seen that the lowest scores (red area) were found in sub-clause 4.3 (understanding the needs and expectations of interested parties), sub-clause 4.5 (bio-risk management system), sub clause 7.2 (competence) and sub-clause 10.3 (continual improvement). In addition, good implementation was achieved on the items with orange and white areas.

According to the scores above, gaps in bio-risk implementation were identified in each laboratory (Table 4). A gap was obtained based on the scores and the total items of ISO 35001:2019 that had not been implemented. In fact, the scores had a similar trend as the items that had not been implemented. In addition, the widest gaps found were those linked to leadership (12.5/50), support (9.5/55), and then performance evaluation (3/23). In the leadership element, items on sub-clause 5.1 had not been complied yet as the top management had not demonstrated leadership and commitment regarding ensuring roles, responsibilities and authorities related to bio-risk management as those had not been defined, documented and communicated to person in charge of biological materials. In addition, sub-clause 5.3 related to organisational roles, responsibilities and authorities was not complied yet. A bio-risk management committee had not documented and reported the terms of reference to top management (5.3.3) and a bio-risk management advisor had not been appointed yet (5.3.4). Regarding the support element (clause 7), the gap was identified in sub-clause 7.1 related to resources. It was clearly found that the organisation had not implemented the alternative controls to encourage workers who contradicted the vaccination program (7.1.1.1). On sub clause 7.2 about competence, the behaviour and management of workers need an improvement, particularly in the assurance of workers. In addition, a gap was also found in sub-clause 7.3 related to awareness of the legal requirements that govern bio-risk management. On performance evaluation of clause 9, the gap was identified on sub-clause 9.2 regarding internal audits. Internal audits and inspection that conforms to this ISO standard had not been performed.

## 4. Discussion

Clinical and medical laboratories have potential exposure to various occupational hazards such as biological agents originating from handling and treatment of specimens. It is crucial that all members and staff are familiar with methods and precaution measures to handle these agents. All laboratories handling clinical and environmental specimens should meet a basic standard or requirement in order to achieve a good implementation of bio-risk management, involving the knowledge of the current biosafety level, the availability of protocols regarding the chain of guardianship, risk identification, and guidelines in safe handling of biological agents [23]. Various tools and guidelines on biosafety and biosecurity have been established and used as global assessment tools, such as WHO bio-risk management (2006), Centre for Disease Control and prevention (CDC) guidelines for safe work practices in human and animal medical diagnostic laboratories (2012), and the Canadian Biosafety Standard (2015) [21]. Recently, ISO has proposed a standard for bio-risk management for laboratories and other related organisation under ISO 35001:2019. Under this guideline, the organisation is required to meet the goals in applying a bio-risk management system through the principle of plan-do-check-act (PDCA) [20]. This study revealed the result of gap analysis of bio-risk management at a National Referral Clinical Laboratory Centre in Indonesia based on this standard. Our findings show that all laboratories in the centre have complied with current standards for bio-risk management. However, the organisation needs improvement on leadership, support, performance evaluation and operation.

Biosafety and biosecurity are important elements of public health systems in addressing the global health issues related to infectious disease. However, the potential hazards and risks need to be considered in handling these elements in laboratory settings. In fact, laboratory-acquired infections (LAIs) can occur in clinical and medical laboratories. This a public health concern as an infected laboratory may prove to be transmission risk to other people or communities [24]. Moreover, the laboratory infections not only affect the health of laboratory staff, but may also cause the accidental leakage of organisms. This potential accident can cause an epidemic in the community near a laboratory and induce a serious global impact on public health. An outbreak in human history has been well documented, for instance laboratory staffs in China were infected to Severe Acute Respiratory Syndrome (SARS) in 2004 spreading new infection to many places [25]. In 2018, the release of biological agents affected the community in Nigeria with Lassa fever, and Nipah virus that infected people in India. In addition, it is expected that infectious disease will remain a significant threat to public health for the foreseeable future [26]. Therefore, the mitigation risks should be consistently implemented and enforced through bio-risk management in all active laboratories.

The objective of bio-risk management is to control and minimize the risk for employees, communities and the environments; to guarantee that standards are in place and implemented consistently; and to establish guidelines for best practices in bio-risk management [27,28]. Our findings show that the institution has established the purpose and mandate of the organisation as well as the boundaries of work that have been defined clearly and communicated to all levels at the organisation. However, the documentation did not fully comply with the ISO 35001: 2019 requirement due to a lack documentation system in the implementation stage. Most of the laboratories in this study have been more concerned towards identifying the bio-risk management issue and maintaining the safety equipment specific to the ISO accreditation criteria [29].

In terms of leadership, it has been identified that the organisation did not ensure the role, responsibilities and authorities in relation to bio-risk management. Furthermore, the bio-risk management committee have not reported nor documented the term of reference to the top management when actually the top management plays a significant role in enforcing safety practices in all aspects of the bio-risk management programs or policies [30], as well as ensuring the organisation could achieve a successful bio-risk management system implementation [31]. For instance, by integrating bio-risk management to the entire facilities with a clear and concise plan on the division of roles and responsibilities among employees, a comprehensive biosafety program implementation can be ensured. Hence, management, awareness, physical security, accountability for materials, information and transport security, personnel reliability, and emergency response should be included in the comprehensive biosafety program [32]. Importantly, the system performs better when all parties recognize the authority of a Biosafety Director, complete each task voluntarily, and remain fully involved in the compliance process [33]. According to a previous study, proactive leaders can enhance the awareness culture at all levels [34].

In the planning element, all items have been implemented. However, the bio-risk management objectives and planning to achieve them, as well as the control plan, need to be addressed by providing documentations. Therefore, the quality objective will meet the criteria of bio-risk management such as consistent, clearly measured, applicable, monitored, communicated and updated bio-risk management [20]. In fact, planning has an integral part in hazard identification, risk assessment and risk control [35]. Thus, each plan should consider the needs and requirements of each facility and type of work in the organisation [36].

In the element of support, the gap was found on the aspect of worker vaccination, behavioural factors and worker management, personnel reliability measures and awareness. Although the laboratory workers are expected to work safely, a conducive laboratory environment should be provided [6]. The organisation also needs to identify and implement alternative strategies to ensure that all the workers are being vaccinated and how to deal with those who refuse vaccination. It is mandatory for health care workers to be vaccinated to minimize the spread of diseases such as yellow fever and tuberculosis [37]. In terms of behavioural factors, the organisation shall provide an individual support and effective management. Another element that was not implemented was the personnel reliability measures. This element is important to ensure whether workers are liable, trustworthy and competent in identifying a biosecurity or bio-risk issues to the organisation. Hence, the awareness of workers in the organisation also needs to be considered [38,39,40]. By ensuring that the laboratory personnel are skilled and competent, organisation can significantly enhance its compliance and performance [41].

What is also important is that the operation system and performance are required to be tested regularly by the organisation. An effective physical protection control is becoming an important element on the national legal framework of bio-risk management among all United Nations (UN) members, including Indonesia [41]. The operation system and equipment should meet the standard operating procedures (SOP) and it has to be ensured that equipment is maintained regularly for safe operation in the laboratory where the safety equipment mostly utilized [42]. In the aspect of emergency response and contingency planning, the scenario of emergency has not been determined for all credible and foreseeable scenarios that will affect the bio-risk management in the organisation. Several emergencies’ scenarios are expected to be considered such as spill, fire and missing biological agents. The institution also needs to consider all the worst possible scenarios in developing an emergency plan to assess the weakness and strengths of the bio-risk management implementation in the organisation [43].

Regarding the evaluation performance, an organisation has continuously examined the performance and effectiveness of its bio-risk management. Nonetheless, it still needs an improvement in documentation during implementation. In addition, an internal audit and management review are also important for ensuring the suitability and adequacy of the bio-risk management implementation since both elements are part of a biosafety laboratory’s accreditation requirements [44]. Furthermore, a management review will provide a gap analysis and information related to the evaluation of bio-risk management [44]. Although there has been improvement in the organisation, there was a slight concern about whether the corrective action is effective to prevent the incidents or non-conformities encountered. Continual improvements need to be determined for enhancing the suitability, adequacy and effectiveness of the bio-risk management system.

Overall, this study plays an integral part in providing information on bio-risk management performance in Indonesia as a developing country. However, there are some limitations in this study. The data were obtained through the checklist based on the ISO 35001:2019, which uses a self-assessment approach. Hence, the information is prone to bias. This study also did not examine the association between other variables, such as safety commitment, number of staff being trained and biosafety levels, and the implementation of biosafety and biosecurity requirements. Thus, further studies are needed to explore this association. The findings from this study will also be less generalizable since the study participants only represent National Clinical and Medical Referral Centre Laboratories, Indonesia, as the sample. Further studies with bigger sample size and more representative study participants are recommended.

## 5. Conclusions

The findings show that most of the laboratories meet the ISO 35001:2019 requirements. However, there are some limitations in leadership support, documentation and evaluation of the existing bio-risk management system. It is crucial to heighten awareness of the need for laboratories to implement a comprehensive bio-risk management system which cover the laboratory biosafety and biosecurity measures. One way to address this would be to create written rules and regulations at the laboratory top management level to show the presence of leadership support. Through these, all laboratories will be required to comply with the available bio-risk policies, rules and regulations. In the case of inexistence of national guidelines, the international standards could be considered such as the Laboratory Bio-risk Management Standard (Chen Workshop Agreement/CWA 15793:2008) and World Health Organization Laboratory Biosafety Manual and Biosecurity Guideline. Furthermore, a documentation system should be established to regularly record the implementation of bio-risk management. This documentation system involves the access, retrieval and use based on risk, storage and preservation, control of changes, the retention and disposition. Hence, it is important to provide a document control procedure and review regularly the documented information. A well-documented implementation would also be beneficial for the evaluation of a bio-risk management system. The laboratory facilities must keep moving to ensure that their workplaces are safe and that they are able to protect the workers, environment, product (diagnostic or research), and biological agents from threats.

## Figures and Tables

**Figure 1 ijerph-18-02308-f001:**
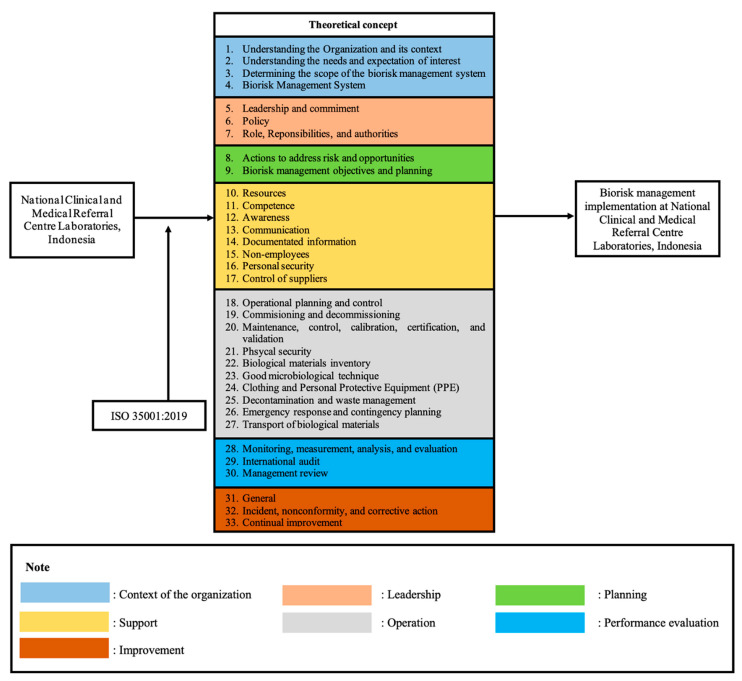
Theoretical concept.

**Figure 2 ijerph-18-02308-f002:**
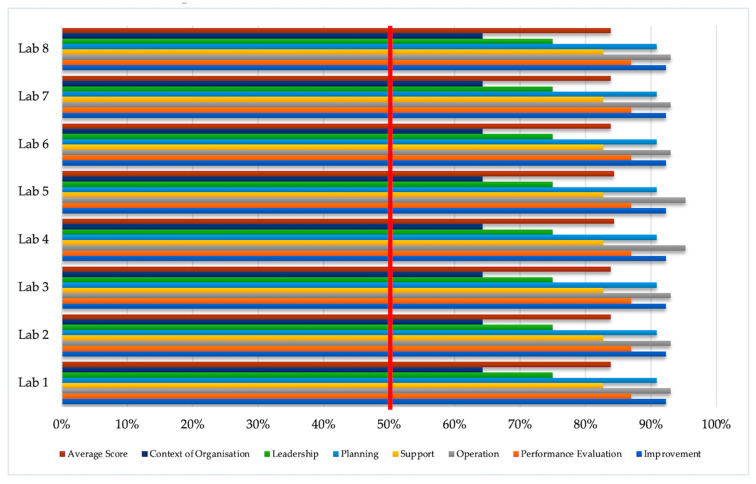
Total score of bio-risk management implementation by element.

**Table 1 ijerph-18-02308-t001:** Eight laboratories of the National Clinical and Medical Referral Centre Laboratories.

Laboratory Name	Type of Laboratory	Type of BSL (Biosafety Level)
Lab 1	Pathology Anatomy	BSL 1
Lab 2	Haematology	BSL 1
Lab 3	Cytogenetic	BSL 1
Lab 4	Molecular	BSL 3
Lab 5	Microbiology	BSL 2
Lab 6	Urinalysis	BSL 1
Lab 7	Automation	BSL 1
Lab 8	Immuno-serum	BSL 1

**Table 2 ijerph-18-02308-t002:** International Standard Organisation (ISO) 35001:2019 elements and checklist development.

No	Element	Sub-Clauses (Number of Items)	Description	**Total of Items**
**1.**	Context and Organisations (Clause 4)	4.1 Understanding the Organisation and its context (2) 4.2 Understanding the needs and expectations of interested parties (2) 4.3 Determining the scope of bio-risk management systems (2) 4.4 Bio-risk management system (1)	The goals and mandates of the organisation, its objectives, and the boundaries of its work must be clearly defined and communicated throughout the organisation. The organisation must determine external and internal issues that are relevant to its objectives and which affect the ability to achieve the expected results of bio-risk management system.	7
**2.**	Leadership (Clause 5)	5.1 Leadership and commitment (10) 5.2 Policy (8) 5.3 Roles, responsibilities, and authorities (32)	Top management must demonstrate leadership and commitment with respect to the bio-risk management system	50
**3.**	Planning (Clause 6)	6.1 Actions to address risks and opportunities (6) 6.2 Bio-risk management objectives and planning to achieve them (5)	Actions are needed to identify, assess, and prioritize bio-risk, implement measures to mitigate bio-risk, integrate these actions into the organisation’s bio-risk management system processes, and evaluate the effectiveness of these measures.	11
**4.**	Support (Clause 7)	7.1 Resources (8) 7.2 Competence (14) 7.3 Awareness (7) 7.4 Communication (6) 7.5 Documented information (14) 7.6 Non-employees (1) 7.7 Personal security (2) 7.8 Control of suppliers (3)	The organisation shall determine and provide the necessary support items for the establishment, implementation, maintenance, evaluation, and continuous improvement of the bio-risk management system.	55
**5.**	Operation (Clause 8)	8.1 Operational planning and control (10) 8.2 Commissioning and decommissioning (2) 8.3 Maintenance, control, calibration, certification, and validation (1) 8.4 Physical security (4) 8.5 Biological materials inventory (3) 8.6 Good microbiological technique (2) 8.7 Clothing and personal protective equipment (PPE) (4) 8.8 Decontamination and waste management (6) 8.9 Emergency response and contingency planning (8) 8.10 Transport biological materials (3)	The organisation shall carry out the operations required for the implementation of the bio-risk management system.	43
**6.**	Performance Evaluation (Clause 9)	9.1 Monitoring, measurement, analysis, and evaluation (7) 9.2 Internal audit (8) 9.3 Management review (8)	The organisation must carry out the necessary performance evaluations to enhance the performance of the bio-risk management system	23
**7.**	Improvement (Clause 10)	10.1 General (1) 10.2 Incident, nonconformity, and corrective action (11) 10.3 Continual improvement (1)	Organisations must determine opportunities for improvement from performance evaluation and implement the necessary actions to achieve the desired results of the bio-risk management system	13
	Total Checklist Items			202

**Table 3 ijerph-18-02308-t003:** Total implementation score (T) by sub-clause.

No	Element	Sub-Clauses		Number of Items	Lab 1	Lab 2	Lab 3	Lab 4	Lab 5	Lab 6	Lab 7	Lab 8
					T (%)	T (%)	T (%)	T (%)	T (%)	T (%)	T (%)	T (%)
**1**	Context and Organisations	4.2	Understanding the Organisation and its context	2	1.5 (75)	1.5 (75)	1.5 (75)	1.5 (75)	1.5 (75)	1.5 (75)	1.5 (75)	1.5 (75)
	(Clause 4)	4.3	Understanding the needs and expectations of interested parties	2	1 (50)	1 (50)	1 (50)	1 (50)	1 (50)	1 (50)	1 (50)	1 (50)
		4.4	Determining the scope of bio-risk management systems	2	1.5 (75)	1.5 (75)	1.5 (75)	1.5 (75)	1.5 (75)	1.5 (75)	1.5 (75)	1.5 (75)
		4.5	Bio-risk management system	1	0.5 (50)	0.5 (50)	0.5 (50)	0.5 (50)	0.5 (50)	0.5 (50)	0.5 (50)	0.5 (50)
**2**	Leadership	5.1	Leadership and commitment	10	8.5 (85)	8.5 (85)	8.5 (85)	8.5 (85)	8.5 (85)	8.5 (85)	8.5 (85)	8.5 (85)
	(Clause 5)	5.2	Policy	8	6.5 (81)	6.5 (81)	6.5 (81)	6.5 (81)	6.5 (81)	6.5 (81)	6.5 (81)	6.5 (81)
		5.3	Roles, responsibilities, and authorities	32	22.5 (70)	22.5 (70)	22.5 (70)	22.5 (70)	22.5 (70)	22.5 (70)	22.5 (70)	22.5 (70)
**3**	Planning	6.1	Actions to address risks and opportunities	6	6 (100)	6 (100)	6 (100)	6 (100)	6 (100)	6 (100)	6 (100)	6 (100)
	(Clause 6)	6.2	Bio-risk management objectives and planning to achieve them	5	4 (80)	4 (80)	4 (80)	4 (80)	4 (80)	4 (80)	4 (80)	4 (80)
**4**	Support	7.1	Resources	8	6 (75)	6 (75)	6 (75)	6 (75)	6 (75)	6 (75)	6 (75)	6 (75)
	(Clause 7)	7.2	Competence	14	8.5 (60.7)	8.5 (60.7)	8.5 (60.7)	8.5 (60.7)	8.5 (60.7)	8.5 (60.7)	8.5 (60.7)	8.5 (60.7)
		7.3	Awareness	7	5 (71.4)	5 (71.4)	5 (71.4)	5 (71.4)	5 (71.4)	5 (71.4)	5 (71.4)	5 (71.4)
		7.4	Communication	6	6 (100)	6 (100)	6 (100)	6 (100)	6 (100)	6 (100)	6 (100)	6 (100)
		7.5	Documented information	14	14 (100)	14 (100)	14 (100)	14 (100)	14 (100)	14 (100)	14 (100)	14 (100)
		7.6	Non-employees	1	1 (100)	1 (100)	1 (100)	1 (100)	1 (100)	1 (100)	1 (100)	1 (100)
		7.7	Personal security	2	2 (100)	2 (100)	2 (100)	2 (100)	2 (100)	2 (100)	2 (100)	2 (100)
		7.8	Control of suppliers	3	3 (100)	3 (100)	3 (100)	3 (100)	3 (100)	3 (100)	3 (100)	3 (100)
**5**	Operation	8.1	Operational planning and control	10	8.5 (85)	8.5 (85)	8.5 (85)	9.5 (85)	9.5 (95)	8.5 (85)	8.5 (85)	8.5 (85)
	(Clause 8)	8.2	Commissioning and decommissioning	2	2 (100)	2 (100)	2 (100)	2 (100)	2 (100)	2 (100)	2 (100)	2 (100)
		8.3	Maintenance, control, calibration, certification, and validation	1	1 (100)	1 (100)	1 (100)	1 (100)	1 (100)	1 (100)	1 (100)	1 (100)
		8.4	Physical security	4	3.5 (87.5)	3.5 (87.5)	3.5 (87.5)	3.5 (87.5)	3.5 (87.5)	3.5 (87.5)	3.5 (87.5)	3.5 (87.5)
		8.5	Biological materials inventory	3	3 (100)	3 (100)	3 (100)	3 (100)	3 (100)	3 (100)	3 (100)	3 (100)
		8.6	Good microbiological technique	2	2 (100)	2 (100)	2 (100)	2 (100)	2 (100)	2 (100)	2 (100)	2 (100)
		8.7	Clothing and personal protective equipment (PPE)	4	4 (87.5)	4 (87.5)	4 (87.5)	4 (87.5)	4 (87.5)	4 (87.5)	4 (87.5)	4 (87.5)
		8.8	Decontamination and waste management	6	6 (100)	6 (100)	6 (100)	6 (100)	6 (100)	6 (100)	6 (100)	6 (100)
		8.9	Emergency response and contingency planning	8	7 (87.5)	7 (87.5)	7 (87.5)	7 (87.5)	7 (87.5)	7 (87.5)	7 (87.5)	7 (87.5)
		8.10	Transport biological materials	3	3 (100)	3 (100)	3 (100)	3 (100)	3 (100)	3 (100)	3 (100)	3 (100)
**6**	Performance Evaluation	9.1	Monitoring, measurement, analysis, and evaluation	7	6 (87.5)	6 (87.5)	6 (87.5)	6 (87.5)	6 (87.5)	6 (87.5)	6 (87.5)	6 (87.5)
	(Clause 9)	9.2	Internal audit	8	7 (87.5)	7 (87.5)	7 (87.5)	7 (87.5)	7 (87.5)	7 (87.5)	7 (87.5)	7 (87.5)
		9.3	Management review	8	7 (87.5)	7 (87.5)	7 (87.5)	7 (87.5)	7 (87.5)	7 (87.5)	7 (87.5)	7 (87.5)
**7**	Improvement	10.1	General	1	1 (100)	1 (100)	1 (100)	1 (100)	1 (100)	1 (100)	1 (100)	1 (100)
	(Clause 10)	10.2	Incident, non-conformity, and corrective action	11	10.5 (95.5)	10.5 (95.5)	10.5 (95.5)	10.5 (95.5)	10.5 (95.5)	10.5 (95.5)	10.5 (95.5)	10.5 (95.5)
		10.3	Continual improvement	2	1 (50)	1 (50)	1 (50)	1 (50)	1 (50)	1 (50)	1 (50)	1 (50)

Red area (lowest score); orange and white area (good implementation).

**Table 4 ijerph-18-02308-t004:** Gap analysis of each laboratory.

No	Element	Lab 1	Lab 2	Lab 3	Lab 4	Lab 5	Lab 6	Lab 7	Lab 8
1.	Context of Organisation	2.5	2.5	2.5	2.5	2.5	2.5	2.5	2.5
2.	Leadership	12.5	12.5	12.5	12.5	12.5	12.5	12.5	12.5
3.	Planning	1.0	1.0	1.0	1.0	1.0	1.0	1.0	1.0
4.	Support	9.5	9.5	9.5	9.5	9.5	9.5	9.5	9.5
5.	Operation	3	3	3	2	2	3	3	3
6.	Performance Evaluation	3	3	3	3	3	3	3	3
7.	Improvement	1	1	1	1	1	1	1	1
Total		32.5	32.5	32.5	31.5	31.5	32.5	32.5	32.5
Effectiveness		83.9%	83.9%	83.9%	84.4%	84.4%	83.9%	83.9%	83.9%

Note. Yellow area refers to the three maximum gap analysis in each laboratory.

## Data Availability

The data presented in this study are available on request from the corresponding author.

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
