# Peer review of "Implementation of Bio-Risk Management System in a National Clinical and Medical Referral Centre Laboratories"

_ijerph, 2021, doi:10.3390/ijerph18052308_

Round 1

Reviewer 1 Report

I feel like the authors have put a lot of effort in their investigation and writing the manuscript. However, the manuscript would need to be revised substantially.

The laboratory containment levels are state clearly including BSL1, 2 and 3. Laboratory containments especially those of the highest levels (BSL3,4) are linked to public health risks. Therefore, results that might have a dual use potential should be excluded from the paper. This public health risk and mitigation of risks should be discussed further. In the material and method sections that should be descriptive, some results (Line 153 for instance) are given. The section should be revised accordingly.

In total, to make it easier to read, I would recommend to focus on essential points e.g. Line 166-169 could be summarized to: In total, six BSL1, one BSL2 and one BSL3 high containment are available (Table 1). Also, only essential Figures should be included, avoiding duplications. I would strongly recommend to review and shorten the discussion.

Specific comments

Line 16-17: Please, could you rephrase the first sentence of the abstract. I would recommend to make two sentences.

Line 17-18: I do not agree to that statement: A comprehensive bio-risk management system is needed to provide a framework for organization to conduct bio-risk assessment. Performing a bio-risk assessment will allow to establish a comprehensive bio-risk management system.

Line 20-21: Please, could you spell out PT and PX, once.

Line 24-25: I would kindly recommend to name an example of what was done best and worst. As well please make clear how many elements were reviewed.

Line 32-34: I would kindly ask you to rephrase the sentence clearly.

Line 35: after the reference [1] a point to finish the sentence is missing.

Line 50: between exposed and by is too much spacing.

Line 60: same comment as for line 50

Line 107-119: I would recommend to shorten this part to a single sentence.

Line 166-169: The lines could be summarized. Like for instance: In total, six BSL1, one BSL2 and one BSL3 high containment are available (Table 1).

Line 174 – 189: I would kindly ask you to summarize the most information in one to two sentences and to refer to Table 2.

Table 1: The head of the table No and laboratory Name is redundant, one column would be enough.

Author Response

Comments and Suggestions for Authors

I feel like the authors have put a lot of effort in their investigation and writing the manuscript. However, the manuscript would need to be revised substantially.

Response General: Dear Reviewer 1

Thank you for your feedback and your time to review our paper. We are very grateful as it helps us to increase the quality of our paper. Herewith, we responded your feedback based on each point on these comments.

Point 1: The laboratory containment levels are state clearly including BSL1, 2 and 3. Laboratory containments especially those of the highest levels (BSL3,4) are linked to public health risks. Therefore, results that might have a dual use potential should be excluded from the paper. This public health risk and mitigation of risks should be discussed further. In the material and method sections that should be descriptive, some results (Line 153 for instance) are given. The section should be revised accordingly.

Response 1: We have excluded the point of (BSL3,4) that are linked to public health risks. We will consider this as our limitation and recommendation for next studies. We agree with your comment that this study is descriptive approach and we have already added into our manuscript. We revised it to “Descriptive method was used in this study to examine the implementation of Bio-risk Management System in National Referral Centre Laboratory, called PX by combining qualitative and quantitative approach, and then gap analyses was also conducted based on the examination findings”

Point 2: In total, to make it easier to read, I would recommend to focus on essential points e.g. Line 166-169 could be summarized to: In total, six BSL1, one BSL2 and one BSL3 high containment are available (Table 1). Also, only essential Figures should be included, avoiding duplications. I would strongly recommend to review and shorten the discussion.

Response 2: It has revised according to recommendation. We also tried to make shorten the part of discussion

Specific comments

Point 3: Line 16-17: Please, could you rephrase the first sentence of the abstract. I would recommend to make two sentences.

Responses 3: It revised  by making two sentences. “The increasing biological threat from biological agents has become concerned in laboratory, and emerging infectious diseases has demanded the awareness and preparedness of laboratory facilities to identify the potential exposures and containment strategies.”

Point 4: Line 17-18: I do not agree to that statement: A comprehensive bio-risk management system is needed to provide a framework for organization to conduct bio-risk assessment. Performing a bio-risk assessment will allow to establish a comprehensive bio-risk management system.

Responses 4: Thank you for correction. It has been revised to “Performing a bio-risk assessment is needed to provide a framework for organization to establish a comprehensive bio-risk management system.”

Point 5: Line 20-21: Please, could you spell out PT and PX, once.

Responses 5: It has been revised by using term PX

Point 6: Line 24-25: I would kindly recommend to name an example of what was done best and worst. As well please make clear how many elements were reviewed.

Responses 6: We changed it to “This study aims to identify the gap between bio-risk management system implementation according to ISO 35001: 2019. 202 items form main seven elements was examined. The findings show that more than half of the elements on ISO 35001:2019 have been implemented. The good performance was identified at lab 4 and 5 which obtained highest scores particularly in element of context of organization, planning, operation, and improvement. Although the widest gap found is linked to leadership, support, followed by performance evaluation.”

Point 7: Line 32-34: I would kindly ask you to rephrase the sentence clearly.

Response 7: Laboratory service facility is functioning to extract biological materials which are handled globally for certain purposes like education, scientific, pharmaceutical and health-related production. The facilities also be potentially exposed with dangerous microorganisms that will harm people and disrupted environment. Change to:

Laboratory service facility has integral part in extracting biological materials that used for certain purposes like education, scientific, pharmaceutical and health-related production. In addition, the laboratory facilities can pose several hazards and risks due to dangerous microorganisms, which can harm people and disrupted environment.

Point 8: Line 35: after the reference [1] a point to finish the sentence is missing.

Response 8: We were missing (.) and it has been conducted re-proofread.

Point 9: Line 50: between exposed and by is too much spacing.

Responses 9: It has been revised by reducing space

Point 10: Line 60: same comment as for line 50

Response 10: It has been revised by reducing space

Point 11: Line 107-119: I would recommend to shorten this part to a single sentence.

Responses 11: We made a shorten to “The centre of national referral laboratory of PX is located in a high-rise building with 10 storeys and one basement, and it offers more than 600 types of laboratory’s assessment with the average services capacity of 130.000 tests each month”

Point 12: Line 166-169: The lines could be summarized. Like for instance: In total, six BSL1, one BSL2 and one BSL3 high containment are available (Table 1).

Responses 12: We revised according the recommendation “Six BSL 1, one BSL 2 and one BSL3 high containment are available (Table 1).

Point 13: Line 174 – 189: I would kindly ask you to summarize the most information in one to two sentences and to refer to Table 2.

Responses 13: it has been revised to “Based on the checklist used, we observed seven main clauses that consist of different items for each clause starting from context of the organizations to improvement (Table 2)”.

Point 14: Table 1: The head of the table No and laboratory Name is redundant, one column would be enough.

Responses 14: We have already revised it.

Reviewer 2 Report

See attached

Author Response

Response to Reviewer 2 Comments

Bio-risk Management Implementation System in National Referral Center Laboratory: An Analysis Study in PT. PX

General

  • Submit manuscript to an editor to assist with phrasing of sentences into readable English. Most sentences throughout the manuscript are not coherent

E.g. Line 34-36: The facilities also be potentially exposed with dangerous microorganisms that will harm people and disrupted environment [1] Laboratories that handle hazardous pathogens must be responsible to manage the safety and security threats 36 released by these biological agents

Thank you for advice. We acknowledge that our manuscript might have been not coherent. But, we have now submitted to editor again for double check and assist the phrasing of sentence.

Methodology

  • Point 1: The title seem to suggest a description/presentation of an “Implementation system”. Is not clear whether the authors want to describe (i) results of a gap analysis (ii) a process/steps that the Lab went through in implementation Biorisk – and then show the results (ii) conduct gap analysis, describe process of corrective action

Responses 1: According to the comment above, we proposed that the main purposes of the manuscript is to assess a gap in implementation of ISO 35001:2019 about bio-risk management for laboratories. Hence we would like to revise the title from “Bio-risk Management Implementation System in National Referral Center Laboratory: An Analysis Study in PT. PX”to “ Implementation of Bio-risk Management System in National Referral Center Laboratory: An Analysis Study in PX”

  • Point 2: The objective of the study (lines 86-88): Hence, this study aims to identify the gap between bio-risk management system implementation according to ISO 35001: 2019 and the existing implementation in the targeted laboratories. There are no results to reflect the second part of the objective

Response 2: We agree that it seems not completed. The main purpose is to identify the gap between bio-risk management system implementation according to ISO 35001: 2019. the existing implementation in the targeted laboratories has been deleted.

  • Point 3: Who conducted the assessment? This allows readers to assess the extend to which the results they present can be used to gauge status of biorisk in the labs

Responses 3: The assessment was by Professor in Occupational Health and Safety and biological expert who certified in bio-risk management system. We added it into manuscript

  • Point 4: It may assist to describe what transpired up to the period of assessment - if any.

Responses 4: not available

Results

  • Point 5: Suggest to use this section to describe results only and not discuss or explain reason for results obtained e.g. Line 199-200

Responses 5: Thank you for suggestion. We deleted this section that explaining the reason.

  • Point 6: Given the high performance of the Labs 1-8 (in overall scores), it is challenging to note the value the results/study adds to the issue of Biorisk Management. I suggest that authors probably highlight the gradual results of each of the sections of the checklist if (i) there are common trends in specific sections that Lab generally do not perform well - and examine why (ii) examine what the good/poor performance could be related (conduct analysis of associations, or correlations) to by collecting data on other variables like number of staff trained in biorisk management, Biosafety Level, staff commitment etc.

Responses 6: Thank you for the feedback. We agree with your suggestion on (giving the high performance of the Labs 1-8 (in overall scores), therefore we have removed the previous figure 2. The results for each section have provided. However, the trend was similar in all laboratories.  The examination of good or poor performance just obtained based on gap analysis found on the checklist items. We are appreciated for your recommendation to conduct analysis of association/correlation. But, we apologize due to the limitation of study, we did not conduct a number of staff trained, staff commitment etc. We will put this aspect to our limitation on this study as consideration for further research.

  • Point 7: Results can be presented to answer the theoretical framework presented by categorizing some of the results as per the framework

Responses 7: We have now updated the results as per the framework and according to per element of ISO 35001:2019. However, the trend was similar in all laboratories and it is not possible to provide graph of each element. Therefore, we put the table 3 in revised manuscript.

  • Point 8: Table 3: It is challenging to understand the message from Table 3. All Labs seem to get same scores across. The numbers/scores what are they related to e.g. the 2.5 - is that a full mark score (100%) or its 2.5 out of possible 10. This may communicate the message better

Response 8: Table 3 shows the gap in bio-risk implementation in each laboratory.  A score based on the total items of ISO 35001:2019 that has not implemented. In fact, the scores was similar as the same items have not complied.  The widest gap found is linked to leadership (12.5/50), support (9,5/55), followed by performance evaluation (3/23).

Conclusion

  • Point 9: Line 337-338: It is crucial to heighten awareness on the need of laboratories to implement a corehensive bio-risk management system which cover the laboratory biosafety and laboratory biosecurity measures – there is need to link the conclusion to results/findings from the study from which this conclusion is based
  • Document a conclusion in 1-2 sentences based on the findings from the study and then add 1-2 sentences to explain – if need be. At the moment the conclusion is not clear

Responses 9: We updated the conclusion section

The findings show that most of the laboratories meet the ISO 35001:2019 requirement. However, there are some limitations in leadership support, documentation, and evaluation on existing bio-risk management system. It is crucial to heighten awareness on the need of laboratories to implement a comprehensive bio-risk management system which cover the laboratory biosafety and laboratory biosecurity measures. One way to address this would be to create written rules and regulations at the laboratory top management level to show leadership support. Through these, all laboratory require to comply to the available bio-risk policies, rules, and regulations. In the case of inexistence of national guidelines, the international standards could be considered. Further, a documentation system should be established to regularly record the implementation of bio-risk management. A well-documented implementation would also be beneficial for the evaluation of bio-risk management system. The laboratory facilities must keep the motion to ensure that they are safe workplaces that can protect the workers, the environment, the product (diagnostic or research) as well as the biological agents.

Reviewer 3 Report

Dear Authors

Abstract includes introductory statement that outlines the background and significance of the study.

The topic addressed in the manuscript treats an important concern in public health.

Introduction summarizes relevant research to provide context and clearly state the problem.  The topics are well developed and confronted to other publications.

Methods are sufficient explained to replicate the research.

The discussion section interprets the findings in view of the results obtained in this and in past studies on this topic.

The conclusions are supported by the research results.

References cited are recent and have a high relevance to the problem.

Minor corrections

Table 1

Replace “Molucelar” with “Molecular”

Replace “Mircrobilogy” with “Microbiology”

Replace “Immunusero” with “immuno-serum”

Author Response

Response to Reviewer 3 Comments

Dear Authors

Abstract includes introductory statement that outlines the background and significance of the study.

The topic addressed in the manuscript treats an important concern in public health.

Introduction summarizes relevant research to provide context and clearly state the problem.  The topics are well developed and confronted to other publications.

Methods are sufficient explained to replicate the research.

The discussion section interprets the findings in view of the results obtained in this and in past studies on this topic.

The conclusions are supported by the research results.

References cited are recent and have a high relevance to the problem.

Minor corrections

Point 1:

Table 1

Replace “Molucelar” with “Molecular”

Replace “Mircrobilogy” with “Microbiology”

Replace “Immunusero” with “immuno-serum”

Response 1:

Dear Sir/Madam

First of all, we would like to say thank you for your time reviewing our paper. All your advices helped us to improve our paper quality.  In accordance with your correction, we have now change these sentences.

Change “Molucelar” to “Molecular”

Change “Mircrobilogy” to “Microbiology”

Change “Immunusero” to “immuno-serum”

Round 2

Reviewer 1 Report

Broad comment

Thank you for revising your manuscript it has certainly improved. Nevertheless, the public health risk and the mitigation of risks were not included into the manuscript. Therefore, I would again kindly ask you to discuss public health risk and mitigation of risks further.

Specific comments

Line 15-17: Please, rephrase the sentences. I would recommend either to delete the part ‘to identify the potential exposures and containment strategies’ or to put it more clearly what you are referring to.

Line 20-21: Please, could you spell out PX, once. What does it stand for?

Line 24-25: I would kindly recommend to state clearly in the result section what was done best and worst.

Line 103-104: Gap analyses was also performed based on the examination findings. I would kindly ask you to change the sentence into: Gap analyses was also performed based on results. In line 161-163 you refer again to gap analysis, perhaps you could consider to move the sentence there.

Line 113-122: Please could you shorten the section.

Line 135: ‘criterion in mine’ could you please explain what you are referring to.

Line 139: Please change the wording from ‘The instrument used’ into ‘the checklist applied’.

Table 2: in the heading you refer to questions, would it be possible the change the wording? Perhaps items would be work?

Line 159-160: this is a result, could you please remove that form the material and method section.

Line 231 – 327: Please, discuss in the section the public health risk and the mitigation of risk by improving/setting up bio-risk management systems.

Line 232: the sentence should be rephrased.

Line 235: You refer to ‘meet a basic standard or requirement’ could you be more specific.

Line 268: You refer to ‘comprehensive biosafety program implementation’, please give an outline, what should be included in the comprehensive biosafety program

Line 337: please, cite international standards.

Line 338: please, give more details about a possible documentation system and make suggestion which one to use

Author Response

Response to Reviewer 1 Comments (Round 2)

Broad comment

Point 1: Thank you for revising your manuscript it has certainly improved. Nevertheless, the public health risk and the mitigation of risks were not included into the manuscript. Therefore, I would again kindly ask you to discuss public health risk and mitigation of risks further.

Response 1: We are very grateful for the recommendation to enhance the quality of manuscript. We have already discussed public health risk and mitigation of risks further in discussion section.

Specific comments

Point 2. Line 15-17: Please, rephrase the sentences. I would recommend either to delete the part ‘to identify the potential exposures and containment strategies’ or to put it more clearly what you are referring to.

Response 2: It has been deleted according to your recommendation.

Point 3: Line 20-21: Please, could you spell out PX, once. What does it stand for?

Response 3:  PX  is a clinical and Medical Referral Centre Laboratories in Indonesia. Due to the confidential issue, we couldn’t mentioned the name of company. Hence , we put on “PX” on this paper. But, we have already updated the title of this paper to avoid the PX that can be misunderstanding. We revised the title to “Implementation of Bio-risk Management System in National Clinical and Medical Referral Centre Laboratories

Point 4: Line 24-25: I would kindly recommend to state clearly in the result section what was done best and worst.

Response 4: We have added the information as your advice “Good performance was identified at lab 4 and 5 which obtained the highest scores, particularly in context of organisation, planning, operation and improvement elements. However, the widest gap was found in leadership, support and performance evaluation”. In addition, we also pointed out in the result. Table 3. illustrates the total score and the percentage of implementation of bio-risk management per sub-clauses. It can be seen that the lowest scores (red area) were found in sub-clause 4.3 (understanding the needs and expectations of interested parties), sub-clause 4.5 (bio-risk management system), sub clause 7.2 (competence) and sub-clause 10.3 (continual improvement). In addition, the good implementation has been achieved on the items with orange and white area.

Point 5: Line 103-104: Gap analyses was also performed based on the examination findings. I would kindly ask you to change the sentence into: Gap analyses was also performed based on results. In line 161-163 you refer again to gap analysis, perhaps you could consider to move the sentence there.

Response 5: We changed the sentence into: Gap analyses was also performed based on results, and line 161-163 was moved according to your advice.

Point 6: Line 113-122: Please could you shorten the section.

Response 6: We have shorten the section as your recommendation. This study was carried out at a National Clinical and Medical Referral Centre Laboratories, Indonesia. This centre is the biggest private laboratory chain which consist of 174 branches, 3 regional referral laboratories, and one national referral laboratory centre across Indonesia. The national referral laboratory centre which is located in Central Jakarta has been established since 2008. This referral laboratory which occupied a high-rise building with 10 stories and a basement, has become the nationwide referral centre for clinical specimen assessment and collects samples from all branches and public hospitals as well as private hospitals in Indonesia. This Centre offers more than 600 types of laboratory’s diagnostic tests with the average services capacity of 130,000 tests each month.

Point 7: Line 135: ‘criterion in mine’ could you please explain what you are referring to.

Response 7: We apologize for that incorrect sentence. We revised ‘criterion in mine” to criteria.

Point 8: Line 139: Please change the wording from ‘The instrument used’ into ‘the checklist applied’.

Response 8: Thank you for your advice, it has been changed to “The checklist applied”

Point 9: Table 2: in the heading you refer to questions, would it be possible the change the wording? Perhaps items would be work?

Response 9: It has been revised to “items” as your review.

Point 10: Line 159-160: this is a result, could you please remove that form the material and method section.

Response 10: we have already removed it according to your advice.

Point 11: Line 231 – 327: Please, discuss in the section the public health risk and the mitigation of risk by improving/setting up bio-risk management systems.

Response 11: We have already discussed your recommendation at discussion section. “Biosafety and biosecurity are important elements of public health system in addressing the global health issues related to infectious disease. However, the potential hazards and risks need to be considered in handling these elements in laboratory settings. In fact, Laboratory-acquired infections (LAIs) can occur in clinical and medical laboratories. It is being a public health concerned as an infected laboratory may prove to be transmission risk to other people or communities[24]. Moreover, the laboratory infections not only affect the health of laboratory staff, but may also cause the accidental leakage of organisms. This potential accident can cause an epidemic in the community nearby laboratory and induce a serious global impact on public health. An outbreak in human history has been well documented, for instance the laboratory staffs in China were infected to Severe Acute Respiratory Syndrome (SARS) in 2004 spreading new infection in many places[25]. In 2018, the release of biological agents affected the community in Nigeria with the case of Lassa Fever, and Nipah virus that infected people in India. In addition, it has expected that infectious disease will remain a significant threat to public health for the foreseeable future[26]. Therefore, the mitigation risks should be consistently implemented and enforced through bio-risk management in all active laboratories”.

Point 12: Line 232: the sentence should be rephrased.

Response 12: We have already rephrased it into “Clinical and medical laboratories have potential exposure to various occupational hazards such as biological agents originated from handling and treatment of specimens”

Point 13:Line 235: You refer to ‘meet a basic standard or requirement’ could you be more specific.

Response 13: After we have already updated the sentence to make more specific. We changed it into All laboratories handling clinical and environmental specimens should meet a basic standard or requirement in order to achieve a good implementation of bio-risk management, involving the knowledge of the current biosafety level, the availability of protocols regarding the chain of guardianship, risk identification, and guidelines in safe handling of biological agents.

Point 14: Line 268: You refer to ‘comprehensive biosafety program implementation’, please give an outline, what should be included in the comprehensive biosafety program

Response 14: We revised and added additional information related to comprehensive biosafety program. The changes was “a comprehensive biosafety program implementation can be ensured. Hence, management, awareness, physical security, accountability for materials, information and transport security, personnel reliability, and emergency response should be included in the comprehensive biosafety program [29]. Importantly, the system performs better when all parties recognize the authority of Biosafety Director, complete each task voluntary and remain fully involved in the compliance process [30]”

Point 15: Line 337: please, cite international standards.

Response 15: We have added the example of international standards, the international standards could be considered such as Laboratory Bio-risk Management Standard (Chen Workshop Agreement/CWA 15793:2008) and World Health Organization Laboratory Biosafety Manual and Biosecurity Guideline.

Point 16: Line 338: please, give more details about a possible documentation system and make suggestion which one to use

Response 16: We have already provided in more detail related to possible a documentation system. This documentation system involves the access, retrieval and use based on risk, storage and preservation, control of changes, the retention and disposition. It is important to provide a document control procedure and review regularly the documented information.
